**Evaluation of cation exchange membrane performance under exposure to high $Hg^0$ and**
**$HgBr_2$ concentrations**
Matthieu B. Miller[1], Sarrah M. Dunham-Cheatham[2], Mae Sexauer Gustin[2], Grant C. Edwards[1,†]
[1]Department of Environmental Sciences, Macquarie University, Sydney, NSW, 2109, Australia
[2]Department of Natural Resources and Environmental Science, University of Nevada, Reno NV, 89557, United
States
[†]Deceased 10 September 2018
*Correspondence to*: Mae Sexauer Gustin mgustin@cabnr.unr.edu
**Abstract**
Reactive mercury (RM), the sum of both gaseous oxidized Hg and particulate bound Hg, is an
important component of the global atmospheric mercury cycle, but measurement currently
depends on un-calibrated operationally-defined methods with large uncertainty and demonstrated
interferences and artifacts. Cation exchange membranes (CEM) provide a promising alternative
methodology for quantification of RM, but method validation and improvements are ongoing.
For the CEM material to be reliable, uptake of gaseous elemental mercury (GEM) must be
negligible under all conditions, and RM compounds must be captured and retained with high
efficiency. In this study, the performance of CEM material under exposure to high
concentrations of GEM ($1.43\times10^6$ to $1.85\times10^6$ pg m$^{-3}$) and reactive gaseous mercury bromide
($HgBr_2 \sim 5000$ pg m$^{-3}$) was explored, using a custom-built mercury vapor permeation system.
Quantification of total permeated Hg was measured via pyrolysis at 600 °C and detection using a
Tekran® 2537A. Permeation tests were conducted for 24 to 72 hours in clean laboratory air, with
absolute humidity levels ranging from 0.1 to 10 g m$^{-3}$ water vapor. GEM uptake by the CEM
material averaged no more than 0.004% of total exposure for all test conditions, which equates to
a non-detectable GEM artifact for typical ambient air sample concentrations. Recovery of $HgBr_2$
on CEM filters was on average 127% compared to calculated total permeated $HgBr_2$ based on
the downstream Tekran® 2537A data. The low $HgBr_2$ breakthrough on the downstream CEMs
(<1%) suggest that the elevated recoveries are more likely related to sub-optimal pyrolyzer
conditions or inefficient collection on the Tekran® 2537A gold traps.

**1   Introduction**
Mercury (Hg) is a persistent environmental contaminant with a significant atmospheric life time,
and the form and chemistry of Hg is an important determinant of its biogeochemical cycling.
Mercury in the atmosphere is found in three forms: gaseous elemental mercury (GEM), gaseous
oxidized mercury (GOM), and particulate bound mercury (PBM). PBM and GOM are often
quantified together as reactive mercury (RM = GOM + PBM); while the term RM dilutes some
specific information regarding the state of GOM in the atmosphere, it removes some uncertainty
as to whether or not PBM contributes to the Hg collected by the CEMs. Atmospheric GEM, at an
average global background concentration of 1 to 2 ng m$^{-3}$, can be reliably measured with
calibrated analytical instruments (Gustin et al., 2015; Slemr et al., 2015). The measurement of
GOM and PBM requires detection at part per quadrillion (pg m$^{-3}$) concentrations, and depends
currently on un-calibrated operationally defined methods with demonstrated interferences and
artifacts, and concomitant large uncertainty (Marusczak et al., 2017; Jaffe et al. 2014; McClure et
al. 2014; Gustin et al. 2013; Lyman et al. 2010). Recent reviews (Zhang et al., 2017; Gustin et
al., 2015) detail the shortcomings, difficulties, developments, and ongoing improvements needed
for atmospheric RM measurements.
One alternative methodology that may provide improved measurement of ambient RM involves
use of cation exchange membranes (CEM). CEM materials have been used to selectively
measure GOM concentrations in ambient air in previous studies (Huang et al., 2017; Marusczak
et al., 2017; Pierce and Gustin, 2017; Huang and Gustin, 2015a; Huang et al., 2013; Sheu and
Mason, 2001; Ebinghaus et al., 1999; Mason et al., 1997; Bloom et al., 1996). Use of CEM type
filters (then manufactured by Gelman Sciences and referred to as "ion exchange membranes")
for this purpose was first documented in the literature in a conference presentation (Bloom et al.,
1996), though these had also been deployed in an earlier field-based international comparative
study of RM measurement techniques in September, 1995 (Ebinghaus et al., 1999). In the
comparative study, one participating lab deployed a series of ion exchange membranes (for
GOM) behind a quartz fiber filter (for PBM) at a sample flow rate of 9 to 10 Lpm, for 24 h
measurements (filter pore sizes were not reported). Results for PBM and GOM were in similar
ranges of 4.5 to 26 pg m$^{-3}$ and 13 to 23 pg m$^{-3}$, respectively (Ebinghaus et al., 1999).
The ion exchange membrane method was also applied in a 1995-96 field campaign for
determining the speciation of atmospheric Hg in the Chesapeake Bay area (Mason et al., 1997).
This study used a 5-stage Teflon filter pack system that included one up front quartz fiber filter
(0.8 μm pore size) to remove particles, and four downstream Gelman ion exchange membranes
(pore size not reported) to 1) capture GOM, 2) capture GOM breakthrough, 3) serve as
deployment blanks, and 4) isolate the filter train on the downstream side (Mason et al., 1997).
Concentrations of GOM were reported to be 5-10 pg m$^{-3}$, essentially at or below the method
detection limit and it was speculated that even this small amount may have been an artifact from
fine particulate Hg passing through the 0.8 μm quartz fiber filter (Mason et al., 1997).  These low
concentrations are likely due to GOM being degraded on the quartz fiber filter or inefficient
uptake by the Gelman filter (see Supplemental Information Gustin et al. 2013). The 3[rd]-in-series
ion exchange membrane blanks were reported to be not significantly different in Hg
concentration from unused membrane material, indicating that breakthrough was not a
phenomenon that extended past the second ion exchange filter position.
The particulate Hg artifact problem was subsequently elaborated on in a further comparative
study focusing exclusively on RM measurement techniques (Sheu and Mason, 2001). Specific
concerns included physical particle breakthrough, re-evolution of gas-phase $Hg^{2+}$ from PBM
captured on the upstream particulate filters passing downstream to the ion exchange membranes,
possible adsorption of GOM compounds to the particulate filters, or a GEM collection artifact on
the ion exchange membranes. None of these concerns were proven or disproven conclusively.
Recent CEM based sampling systems typically deploy a pair of CEM disc filters without a pre-
particulate filter, in replicates of 2 to 3 at a flow rate of 1.0 Lpm (Gustin et al., 2016). Each pair
of filters constitutes one sample, the first filter serving as the primary RM collection surface, and
the second filter capturing breakthrough. Filters are deployed for 1 to 2 weeks and then collected
for analysis (Huang et al., 2017). The CEM material consists of a negatively charged
polyethersulfone coated matrix (Pall Corporation), and at least one manufacturing evolution has
occurred (Huang and Gustin, 2015b). Prior CEM material versions (I.C.E. 450) had a pore size
of 0.45 μm, while the current CEM material (Mustang® S) has a manufacturer reported pore size
of 0.8 μm.
Previous work with the I.C.E 450 material indicated it does not adsorb significant quantities of
GEM in passive exposures, but selectively uptakes gas-phase $Hg^{2+}$ species (Lyman et al., 2007).
The CEM material was subsequently adapted for use in active sample flow systems, with the
presumption of continued inertness to GEM and selectivity for GOM (Huang and Gustin, 2015a;
Huang et al., 2013). These studies and others (Lyman et al., 2016) have shown better GOM
recovery on CEM material compared to potassium chloride (KCl) coated denuder methods.
Despite these tests, the transparency of the CEM material to GEM uptake has not been
conclusively demonstrated for active sampling flow rates, nor for high GEM concentrations,
though limited data using low concentration manual $Hg^0$ injections through CEM filters suggests
little or no GEM uptake (Lyman et al., 2016). However, even small rates of GEM uptake by the
CEM material could result in a significant measurement artifact (e.g. a modest 1 to 2% GEM
uptake could easily overwhelm detection of typical ambient GOM concentrations). It is therefore
important that a GEM artifact be ruled out if the CEM material is to be successfully deployed for
ambient RM measurements.
Additionally, previous studies observed significant amounts of "breakthrough" GOM on the
secondary filter. The amount of breakthrough is not consistent, neither as a constant mass, with
total Hg ranging from zero to as high as 400 pg (Huang et al., 2017),  nor as a percentage of Hg
collected on the primary filter, ranging from 0 to 40% (Pierce and Gustin, 2017). Similar variable
breakthrough issues were observed in the earliest field-based CEM measurements as well
(Mason et al., 1997). In contrast to ambient measurements, previous laboratory experiments have
reported only minor (0 to 16%) or no breakthrough  Huang and Gustin, 2015a; Huang et al.,
2013). Limited experimental work with flow rates of 1.0 and 16.7 Lpm in ambient air could not
provide an explanation for differing breakthrough rates (Pierce and Gustin, 2017).
In this research we investigated the potential for GEM uptake on CEM material using a custom-
built permeation system. Tests were done to investigate the ability of a pyrolyzer to convert
GEM to GOM.   In addition, the ability of the CEM material to capture and retain a
representative GOM compound (mercury(II) bromide, $HgBr_2$) was explored, and the collection
efficiency for this compound was estimated. We attempted to explain or rule out possible
mechanisms of RM breakthrough for both dry and humid conditions.



## 2   Methods

### 2.1 System for sampling configuration

A Tekran® 2537A ambient mercury analyzer was integrated with a custom-built permeation system designed to enable controlled exposures of GEM and GOM to CEM filters (Fig. 1). The 2537A analyzer was calibrated at the beginning and periodically throughout the study and checked for accuracy by manual $Hg^0$ injections (mean recovery $101.1\% \pm 4.3$, n = 10, SI Fig. 1). The entire system was checked for Hg contamination in clean air prior to permeation tests, and periodically during sampling (SI Fig. 2). See SI for additional information on Tekran quality control.  All tubing and connections used in the permeation system were polytetrafluoroethylene (PTFE), except for the quartz glass pyrolyzer tube and perfluoroalkoxy (PFA) filter holders. Given its reactive nature, some GOM inevitably adsorbs to internal line surfaces, but the capacity of these materials to sorb and retain GOM is not infinite and a steady state of adsorption/desorption is expected after 5-6 hours of exposure to a stable concentration (Xiao et al., 1997; Gustin et al., 2013).

Sample flow through the system was alternated between two PTFE sample lines (designated Line 0 and Line 1) using a Tekran® Automated Dual Switching (TADS) unit. Sample air was constantly pulled through each line at 1.0 Lpm by the internal pump and mass flow controller (MFC) in the 2537A, or by an external flush pump (KNF Laboport® N86 KNP) and MFC (Sierra Smart-Trak® 2). Laboratory air was pulled through a single inlet at the combined rate of 2.0 Lpm, passing through a 0.2 µm PTFE particulate filter and an activated charcoal scrubber (granular activated carbon 6-12 mesh, FisherChemical®) to produce clean sample air. Additionally, for dry air permeations sample air was pulled through a Tekran® 1102 Air Dryer installed upstream of the particulate filter, and for elevated humidity permeations sample air was

pulled through the headspace of a distilled water bath (DIW, < 0.2 ng L$^{-1}$ total Hg) that was
located upstream from the charcoal scrubber to eliminate the DIW being a potential Hg source to
the system. Temperature and relative humidity (RH) were measured in-line (Campbell Scientific
CS215) and used for calculation of absolute humidity.
Pure liquid $Hg^0$ and crystalline $HgBr_2$ (purity > 99.998% Sigma-Aldrich®) were used as Hg
vapor sources. The elemental $Hg^0$ bead was contained in a PTFE vial. Solid $HgBr_2$ crystals were
packed in thin-walled PTFE heat-shrink tubing (O.D. 0.635 cm) with solid Teflon plugs in both
ends to create a permeation tube with an active permeation length of 2 mm (Huang et al., 2013).
The $HgBr_2$ permeation tube was also placed in the bottom of a PTFE vial, and the permeation
vials were submerged in a temperature-controlled laboratory chiller (0.06 ± 0.13 °C, Cole Parmer
Polystat®). A low source temperature was favored, because higher temperatures would have
produced unacceptably high concentrations, and there is evidence that at higher temperatures a
small amount of $Hg^0$ can be evolved from $Hg^{2+}$ compounds (Xiao et al., 1997).
An ultra-high purity nitrogen ($N_2$) carrier gas was passed through the permeation vials at 0.2
Lpm to carry the target Hg vapor into the main sample line through a PTFE T-junction. The main
sample line was split into Line 0 and Line 1 immediately downstream from the permeation flow
junction, with flow on each line controlled by MFC. Line 0 proceeded directly to the 2537A
without modification during GEM permeations (Fig. 1A), but housed CEM filters during the
$HgBr_2$ permeations (Fig. 1B, 1C). Line 1 contained an in-line pyrolyzer unit. The goal of the
pyrolyzer was to convert all Hg to GEM for detection on the Tekran® 2537A.

### 2.2 Pyrolyzer

The pyrolyzer used in the study (SI Fig. 3) consisted of a 25.4 cm long quartz glass tube of 0.625
cm diameter (custom, URG Corporation). A loosely packed 3 cm section of quartz wool was
lodged in the mid-section of the tube, and this 3 cm section was wrapped with 22 gauge
Nichrome wire (18 loops). The quartz tube was closely contained within 2.5 cm thick quartz
fiber insulation within a 1.6 mm aluminum casing, except for an enclosed air space around the
heated Nichrome coil section. The coil wire was connected to 16 AWG stranded copper wire
with all metal disconnects that were buried within the quartz fiber insulation to reduce thermal
fatigue on the connections. The copper wire insulation was stripped and replaced with higher
temperature heat-shrink insulation where the wiring passed through the pyrolyzer case to the
external power supply. The tip of a 150 mm long K-type thermocouple (Auber WRNK-191) was
inserted through the insulation into the heated air space next to the coil to provide a temperature
feedback for a PID controller (Auber SYL-1512A). Power to the Nichrome coil was supplied by
a 12 VDC transformer through a solid-state relay (Auber MGR-1D4825) switched by the PID
controller. It was found that the position of the feedback thermocouple in the airspace outside of
the heating coil caused a large discrepancy between nominal temperature setpoint and actual
temperature inside the heated section of pyrolyzer tube. In general, much higher temperatures are
achieved inside the coil than outside. To compensate for this, actual temperature at the heated
coil section was verified to 600°C by external IR sensor and internal thermocouple probe.
To test if higher pyrolyzer temperatures converted more GOM to GEM for detection by the
Tekran 2537, the pyrolyzer temperature was increased to 650, 800, and 1,000°C (SI Fig. 4).
Pyrolyzer temperatures were measured by placing a thermocouple inside the pyrolyzer.  GOM
concentrations measured as GEM by the Tekran 2537A increased at 600 and 800°C relative to
375°C. There was no significant difference between the amount of mercury concentrations in the
downstream Tekran 2537A when the pyrolyzer was at 600 and 800°C C (*t-test, p = 0.08*),
indicating that the increased pyrolyzer temperature did not convert more GOM to GEM.
However, when the pyrolyzer temperature was increased to 1000 °C, significantly more mercury
was measured by the downstream Tekran 2537A relative to when the pyrolyzer was at 650°C ($t$-
*test, p = 0.00*), indicating that the higher temperature was more efficient at converting GOM to
GEM; however, the pyrolyzer design could not sustain the 1000 °C temperature and was deemed
unsafe to use in the experimental permeation system. Thus, all experiments were performed with
a pyrolyzer temperature of 600°C.
The residence time in the pyrolyzer tube was approximately 1.5 seconds. Quartz wool was added
to increase the amount of surface area available to facilitate reactions and maximize the amount
of GOM converted to GEM in the pyrolyzer. Based on supplemental experiments, the
downstream Tekran® 2537A Hg measurements when the pyrolyzer was at 650 °C was 75%
compared to the measurements when the pyrolyzer was at 1000 °C (SI Fig. 4), indicating a
higher GOM to GEM conversion efficiency with higher pyrolyzer temperatures. Though this
conversion efficiency value does not describe the exact inefficiency of the experimental system
in this study, it provides an estimate for the efficiency of the pyrolyzer design in this study.
Having an efficient pyrolyzer provides us with a means of constraining permeation tube
permeation rates.
**2.3 Sample deployment**
CEM filters were deployed in 2-stage, 47 mm disc PFA filter holders (Savillex©). The primary
"A" filter in the 2-stage holder is the first to be exposed to the permeated Hg, with the secondary
"B" filter mounted immediately behind the A filter (A to B distance ~ 3mm) to measure potential
breakthrough. For GEM permeations, three 2-stage filter holders were placed in-series on Line 1
behind the pyrolyzer unit (Fig. 1A), while total Hg coming through the system was measured on
Line 0 with no filters in place. This allowed simultaneous exposure of 6 CEM filters in one GEM
sample exposure. The first CEM filter in-line served to scrub any small residual RM passing
through the system and pyrolyzer, and these first in-line filters were removed for the calculations
of mean GEM uptake rate, (SI. 5 and discussion). A controlled experiment was also performed to
ensure that both Lines 0 and 1 were conducting comparable concentrations of mercury under the
experimental conditions. Two-stage filter packs were deployed with CEM filters in each line at
equal distances from the permeation tube. The membranes were deployed for the same amount
of time in triplicate and analyzed to quantify the amount of total mercury sorbed to the
membranes. The average % deviation between lines was 2.9%, with a maximum deviation of
5.4%. These results indicated that though there may be some difference in the amount of
mercury passing through Lines 0 and 1, the difference was relatively small.
For determining the potential for GOM breakthrough, two system configurations were used. In
the first configuration (Fig. 1B), the total Hg concentration of air that passed through the
pyrolyzer on Line 1 was measured without any filters, while Line 0 held one 2-stage CEM filter
pair for $HgBr_2$ loading. This configuration allowed for 10 min interval quantification of the
$HgBr_2$ permeation concentration through Line 1 using the 2537A, and comparison with total Hg
loading on the CEM filters on Line 0.
In the second configuration, replicate filters were concurrently loaded with $HgBr_2$ by placing 2-
stage CEM filter holders on both Line 0 and Line 1 (upstream of the pyrolyzer, Fig. 1C). In all
$HgBr_2$ exposures, the filter holders were placed as close to the permeation vial as possible, with a
total distance from vial to filter surface of approximately 20 cm. Mercury bromide permeation
was conducted in dry air and elevated humidity air. The difference between one line being fully
open to the $HgBr_2$ permeation flow (configuration Fig. 1B) and then closed by deployment of the
CEM filters (configuration Fig. 1C) enabled a rough determination of the amount of $HgBr_2$ line-
loss within the system.
**2.4 Analyses of cation exchange membranes**
After permeation, CEM filters were collected into clean, sterile polypropylene vials and analyzed
for total Hg by digestion in an oxidizing acid solution, reduction to $Hg^0$, gold amalgamation, and
final quantification by cold vapor atomic fluorescence spectrometry (CVAFS, EPA Method
1631, Rev. E) using a Tekran® 2600 system. The system background Hg signal was determined
for every analytical run by analyzing pure reagent solution in the same vials and at the same
volume as used for actual filter samples. Total Hg standards (5 to 100 ppb) were analyzed before
and after each batch of 10 filter samples to check precision and recovery, and the mean recovery
for all Hg standards was $97.2 \pm 5.0$ % (n = 37). Analysis for total Hg on the CEM filters
provided for comparison of total Hg filter loading, and verification of in-line results. A to B filter
breakthrough was calculated by comparison of total Hg recoveries on the primary and secondary
CEM filters, using Eq. (1):
$$\% \, Breakthrough = 100 * CEM_{2nd}/(CEM_{1st} + CEM_{2nd}) \, (1)$$
Blank CEM filters were collected and analyzed in the same manner with every set of sample
filters deployed on the permeation system, and the overall mean filter blank value was subtracted
from all total Hg values to calculate the final blank-corrected Hg values used for data analysis.
All data were analyzed in Microsoft® Excel (version 16.12) and RStudio® (version 3.2.2).

## 3   Results


### 3.1 Elemental Mercury Uptake on CEM Filters


Elemental Hg uptake on CEM material was negligible for permeated $Hg^0$ vapor concentrations
ranging from $1.43 \times 10^6$ to $1.85 \times 10^6$ pg m$^{-3}$ (Fig. 2). High GEM concentrations were employed in
this study under the logic that if no GEM uptake was observed at high concentrations, a similar
lack of GEM uptake can be expected for lower concentrations.
The mean Hg mass on blank CEM filters was $50 \pm 20$ pg (n = 28). For permeations into dry
sample air of $0.5 \pm 0.1$ g m$^{-3}$ water vapor (WV), total mean Hg$^0$ permeation exposures of $2.7 \times 10^6$
pg (24 h) and $7.3 \times 10^6$ pg (72 h) resulted in total (blank-corrected) Hg recoveries on the CEM
filters of $100 \pm 40$ pg (n = 10) and $280 \pm 110$ pg (n = 5), respectively. These quantities of total
recovered Hg equate to a mean GEM uptake rate on the CEM filters of $0.004 \pm 0.002\%$ ($0.006 \pm$
$0.006\%$ including first in-line filter). For GEM permeations into ambient humidity sample air (2
to 4 g m$^{-3}$ WV), at a slightly lower total mean permeated Hg$^0$ 24 h exposure of $2.1 \times 10^6$ pg, total
(blank-corrected) Hg recoveries on the CEM filters were $55 \pm 30$ pg (n =10), equating to a GEM
uptake rate of $0.003 \pm 0.001\%$ ($0.005 \pm 0.005\%$ including first in-line filter).
The first CEM filter in-line during the GEM permeations always showed more total Hg than the
following 5 downstream filters that were not significantly different from each other (SI Fig. 5). It
is unlikely that the Hg observed on the first CEM filters resulted from GEM uptake. Even at the
highest GEM permeation rate, the first filter captured only ~1700 pg of Hg, out of a total
permeated amount of over 7.3 *million* pg (a 0.02 % uptake rate). This means that the downstream
CEM filters were still exposed to about 7.2985 *million* pg of GEM but captured less total Hg. As
we cannot entirely rule out the possibility of some small rate of *in-situ* oxidation of GEM in the
system, at the surface of the Hg$^0$ bead or in the vapor phase, the first in-line filters were not
included in the calculation of GEM uptake rates because of suspicion that some component of
the Hg captured on the first filter was GOM. The overall GEM uptake rate was linear ($r^2 = 0.69$,
p = 0.00; SI Fig. 5) for the range of concentrations used in this study, indicating a similar low
uptake rate can be expected down to lower GEM concentrations. GEM uptake by first in-line
filters was also linear ($r^2$ = 0.92, p = 0.00), though these results are based on 5 data points.
Exponential models poorly fit the experimental data for both the overall GEM uptake and uptake
by first in-line filters (SI Fig. 5).

## 3.2 Mercury Bromide Uptake on CEM Filters

Breakthrough of $HgBr_2$ vapor from the primary (A) to secondary (B) CEM filters was low for all
conditions tested in this study (Table 1). These conditions included $HgBr_2$ permeated into clean
dry laboratory air with < 0.5 g m$^{-3}$ WV, clean air at ambient room humidity (4 to 5 g m$^{-3}$ WV),
and clean air at elevated humidity (10 to 11 g m$^{-3}$ WV), at line temperatures between 17 to 19
°C. Overall, the mean A to B filter breakthrough ranged from 0 to 0.5%, and averaged 0.2 ± 0.2
% (n = 17), with no statistical difference observed in mean breakthrough rates for the three levels
of humidity (ANOVA, p = 0.124).
The first $HgBr_2$ permeation in clean dry (< 0.5 g m$^{-3}$ WV) laboratory air was over a 96 h period,
using the system configuration in Fig. 1B to establish an approximate permeation rate (Fig. 3).
Total Hg reaching the 2537A through the pyrolyzer on Line 1 (red line, Fig. 3) indicated an
average $HgBr_2$ exposure concentration of 4540 pg m$^{-3}$, or about 4.5 pg min$^{-1}$ from the permeation
tube. This permeated concentration of $HgBr_2$ was deliberately much higher than ambient in order
to test retention and break through at high levels. It should be noted that these concentrations are
50 – 1000 times above background ambient concentrations and the performance of the CEM
filters at low concentrations could be slightly different. After this permeation, total blank-
corrected $HgBr_2$ loading on the primary CEM filter on Line 0 was 49400 pg, but only 50 pg on
the secondary CEM filter, indicating a breakthrough rate of approximately 0.1%. Total Hg
reaching the 2537A through the CEM filters on Line 0 (black line, Fig. 3) over this time period
was 15 pg, mostly at the beginning of the deployment when some ambient Hg entered the
opened system. The low concentrations of Hg measured downstream in Line 0 on the 2537A
corroborates that breakthrough of $HgBr_2$ was low. These data also demonstrate that the CEM
material did not saturate with a $HgBr_2$ loading of ~ 50000 pg, a loading far higher than could be
expected in ambient conditions.
Subsequent replicate 24 h $HgBr_2$ permeations in clean dry air resulted in consistent total Hg
loading on CEM filters placed on both lines concurrently ($8560 \pm 320$ pg, n = 6, Samples 2-7
Table 1), and mean total Hg on the secondary CEM filters was $20 \pm 10$ pg (average
breakthrough of 0.3%). On Line 0 (black line, Fig.3), which was never open to $HgBr_2$ vapor
downstream from the CEM filters at any point in the study, Hg measured at the 2537A was zero
for all three 24 h permeations, indicating no breakthrough (Samples 2, 4, & 6, Table 1).
However, on Line 1 that had been exposed to the full $HgBr_2$ vapor concentration of 4540 pg $m^{-3}$
over the duration of the 96 h permeation test, 1155 pg of Hg were measured downstream in the
first 24 h sample (Sample 3, Table 1). The amount of downstream Hg dropped to 10 pg in the
second 24 h, and 6 pg in the third 24 h (Samples 5 & 7, Table 1). This downstream Hg in Line 1
(compared to the zero Hg simultaneously observed on Line 0) is attributed to volatilization of
$HgBr_2$ that had adsorbed to the line material during the open permeation flow. At the moment
CEM filters were deployed on Line 1 (red-to-blue transition, Fig. 3), a rapid asymptotic decline
in the Hg signal began. This decay curve supports drawdown and depletion of a Hg reservoir on
the interior line surfaces behind the CEM filters, and not a continuous source such as
breakthrough from the permeation tube that was still supplying $HgBr_2$ to both sample lines. The
total mass of Hg volatized from the interior line surfaces (1155 pg) represents 4 to 5% of the
total $HgBr_2$ that had passed through Line 1 (~25000 pg based on 2537A measurement).
Eventually, Hg reaching the 2537A through Line 1 decreased to zero during the same 24 h filter
deployment, indicating the majority of $HgBr_2$ line contamination in a high-concentration
permeation system can be expected to flush out within ~12 h. However, we caution that
materials used in high-concentration permeation systems, despite being flushed out, should not
be used for background ambient air work without at least a very thorough acid cleaning.
Additional $HgBr_2$ permeations were made at two levels of in-line humidity. At ambient room
humidity (4 to 5 g m$^{-3}$ WV), mean total Hg measured on the CEM filters was $7910 \pm 520$ pg (n =
4; Samples H2-5, Table 1), with an average breakthrough to the secondary filters of 0.3%. When
normalized for sample volume, the mean $HgBr_2$ loading on CEM filters during ambient humidity
($5968 \pm 125$ pg) and dry air ($5995 \pm 188$ pg) permeations was not statistically significantly
different (t-test $p = 0.790$).  $HgBr_2$ breakthrough rates were also the same (0.3%) as during the
dry air permeations, indicating that the permeation system was operating similarly at the two
humidity levels, and suggesting that absolute humidity concentrations of 4 to 5 g m$^{-3}$ WV had
insignificant effects on collection of $HgBr_2$ in clean laboratory air by the CEM material.
An increase in humidity resulted in an initial large increase in Hg measured at the 2537A
downstream of the CEM filters on Line 0 (Sample H1, Table 1), concurrently with an open
$HgBr_2$ permeation flow through Line 1 while both lines were subjected to increased RH. This
downstream Hg on Line 0 dropped substantially to zero in ~10 h in the first 24 h deployment
(Sample H2, Table 1), and was zero for the duration of the second 24 h deployment (Sample H4,
Table 1). Hg rapidly declined to zero, due to off-gassing from the tubing induced by the
increased humidity, which facilitated a heterogeneous surface reduction of $HgBr_2$ to GEM in the
short section of line between the permeation source and CEM filters. This phenomenon was also
observed during the Reno Atmospheric Mercury Intercomparison eXperiment (RAMIX; Gustin
et al., 2013). Reduced $HgBr_2$ then then passed through to the 2537A as GEM. As the
breakthrough rate and the mean $HgBr_2$ loading on the CEM filters did not change between the
dry air and ambient humidity permeations, the downstream Hg observed at the 2537A during the
ambient humidity permeations cannot be attributed to a loss of Hg from the CEM filters and is
more likely due to a process in the sample lines.
As a further test of possible humidity effects, two replicate 24 h CEM filter deployments were
conducted in elevated humidity conditions (10 to 11 g m$^{-3}$ WV) created by an in-line water bath.
Mean total Hg loading on the primary CEM filters was higher compared to the previous
permeations (11700 ± 720 pg, n = 4, Samples H9-12, Table 1), indicating an increase in the
effective HgBr$_2$ permeation rate, possibly due to the perturbation caused by a poor filter seal and
small leak in the preceding deployment (Sample H7-8, Table 1). However, mean total Hg on the
secondary CEM filters was 20 ± 20 pg, indicating an average breakthrough of 0.1%, less than the
breakthrough observed for the lower humidity permeations.

**4    Conclusions**
GEM uptake on the CEM material was negligible under the laboratory conditions and high GEM
loading rates (3 orders of magnitude above ambient) tested in this study, with an overall linear
uptake rate of 0.004% (SI Fig. 5). This uptake rate would be insignificant at typical ambient
atmospheric Hg concentrations (1 to 2 ng m$^{-3}$). As a hypothetical example, a CEM filter
sampling ambient air at an average GEM concentration of 2 ng m$^{-3}$ for a typical 2-week sample
period would have a total Hg$^0$ exposure of ~40000 pg. At the calculated uptake rate of 0.004%, a
maximum 1.6 pg of Hg observed on the sample filter could be attributed to GEM artifact.
Although further work is required to more definitively determine detection and quantification
limits of the various CEM methodologies, based on the mean total Hg mass of 50 ± 20 pg
observed in this study, the artifact of GEM uptake by the CEMs would be below the detection
limit observed here. This corroborates the lack of GEM uptake seen by Lyman et al. (2016) for
manual $Hg^0$ injections on CEM filters at lower total mass loadings of 300 to 6000 pg.
Mean $HgBr_2$ breakthrough from primary to secondary CEM filters averaged $0.2 \pm 0.2\%$ over all
test conditions. A to B filter breakthrough was derived from a comparison between the large
amount of $HgBr_2$ permeated onto the primary CEM filters, to the small amount of $HgBr_2$ that
collected on the secondary CEM filters, 3 mm immediately downstream. The measurement of
1000s of pg of Hg on the primary filter, and only 10s of pg on the secondary filter, leads to the
conclusion that the primary filter removed the majority of $HgBr_2$ from the sample air stream
under laboratory conditions applied in this study. In addition, low breakthrough was corroborated
by downstream measurement of the air stream passing through the CEM filters, using the
Tekran® 2537A. The average breakthrough to the 2537A was 0 pg for 24 h permeations in dry
air, and 0 to 40 pg in humid air, for filter deployments at steady-state (> 24 h without large
perturbations).
While the permeation system was not specifically optimized for a quantitative mass balance
between permeated $HgBr_2$ and $HgBr_2$ recovered on the CEM filters, a rough estimation of the
CEM collection efficiency is possible. Using the $HgBr_2$ permeations conducted in clean dry air
(mean loading 8560 pg), and comparing this to the mean Hg concentration measured at the
2537A analyzer during the last 24 h of the 96 h permeation measurement (4680 pg m$^{-3}$ or 6739
pg per 24 h), $HgBr_2$ recovery on the CEM filters averaged 127%. Adjusting the expected
permeated $HgBr_2$ mass for our estimated line-loss (~4-5%) changed the recoveries to ~123%.
Still, $HgBr_2$ loading on the CEM filters was ~23% higher than expected based on the pyrolyzed
total measurement on the 2537A, indicating not all $HgBr_2$ was converted to GEM. This can be
explained by the pyrolyzer design used in this study not being 100% efficient at thermally
reducing $HgBr_2$ to $Hg^0$, based on the higher total Hg recoveries on the CEM filters versus total
Hg measured through the pyrolyzer on the Tekran 2537.
The technique of gold amalgamation in general, and specifically including the Tekran® 2537A
analyzer, is widely considered to provide a quantitative total gaseous Hg measurement, at or very
near 100% collection efficiency for $Hg^0$ and Hg compounds (Temme et al., 2003; Landis et al.,
2002; Schroeder et al, 1995; Dumarey et al., 1985; Schroeder and Jackson, 1985). However, to
our knowledge collection and desorption efficiencies on gold traps have not been demonstrated
for $HgBr_2$. The stated desorption temperature of the Tekran® 2537A gold traps is 500 °C, but
temperatures as low as 375 °C have been reported (Gustin et al., 2013). This would cause
reduced thermal decomposition efficiency for all captured GOM compounds, including $HgBr_2$.
We speculate that a combination of incomplete thermal decomposition to $Hg^0$ at both the 600 °C
pyrolyzer and during the best-case 500 °C desorption of the 2537A gold traps contributed to the
~20% non-detection of total permeated $HgBr_2$ as it passed through the CVAFS optical path.
Evidence for this conclusion can be seen in the 134% increase in Hg collection by the Tekran®
2537A when the pyrolyzer was at 1000 °C, as compared to at 650 °C (SI Fig. 4), in
supplementary experiments.
While our results validated some basic performance metrics for the CEM material, they did not
provide data that could fully explain the higher levels of breakthrough observed for CEM filters
deployed in ambient air over the 1-to-2 week sample periods in previous studies. Increasing
humidity by itself did not affect observed $HgBr_2$ breakthrough. A $HgBr_2$ loading of ~50000 pg
also did not lead to increased breakthrough, indicating there is no saturation effect on CEM filter
capacity at a GOM loading far greater than expected from ambient concentrations. It remains
unclear, though, whether breakthrough results from different collection efficiencies for GOM
compounds other than $HgBr_2$, or whether breakthrough results from a degradation of GOM
retention capacity in the CEM material when exposed to ambient air chemistries not simulated in
this study. Also, our experiments were conducted in particulate-free air, which leaves open the
possibility that breakthrough is related to capture (or lack thereof) of PBM by the CEM material.
Further testing and refinements are necessary, beginning with optimization of the pyrolyzer
parameters (e.g., temperature, volume) to allow for a more accurate quantitative comparison
between the CEM and Tekran® 2537A results. Permeation rates of $HgBr_2$ were variable and need
to be more precisely controlled, a standardized and stable GOM permeation system being needed
in general. This study was undertaken using controlled laboratory conditions, but CEM
performance needs to be further validated in ambient air. Specifically, the reasons for RM
breaking through CEM filters deployed in ambient air still need to be determined.
**Acknowledgements**
The authors would like to acknowledge funding from Macquarie University iMQRES 2015148
and NSF Grant 629679. Valuable input and assistance were received from Dr. Ashley Pierce, Dr.
Seth Lyman, and the students of Dr. Gustin's laboratory. We bid an untimely farewell to Dr.
Grant C. Edwards, who was ever a cheerful friend, mentor, and colleague. Dr. Edwards passed
away unexpectedly on September 10, 2018. We thank the diligent reviewers for their valuable
and constructive comments.

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

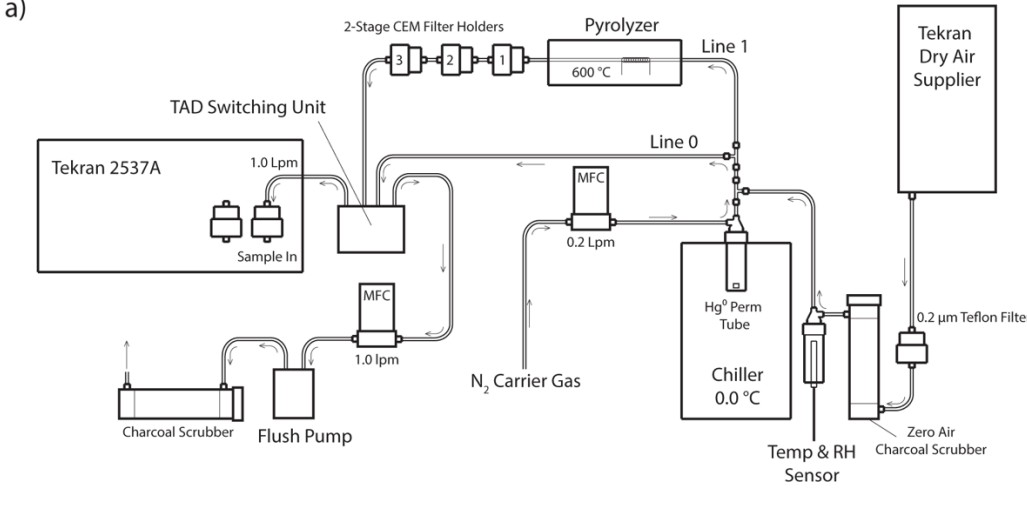

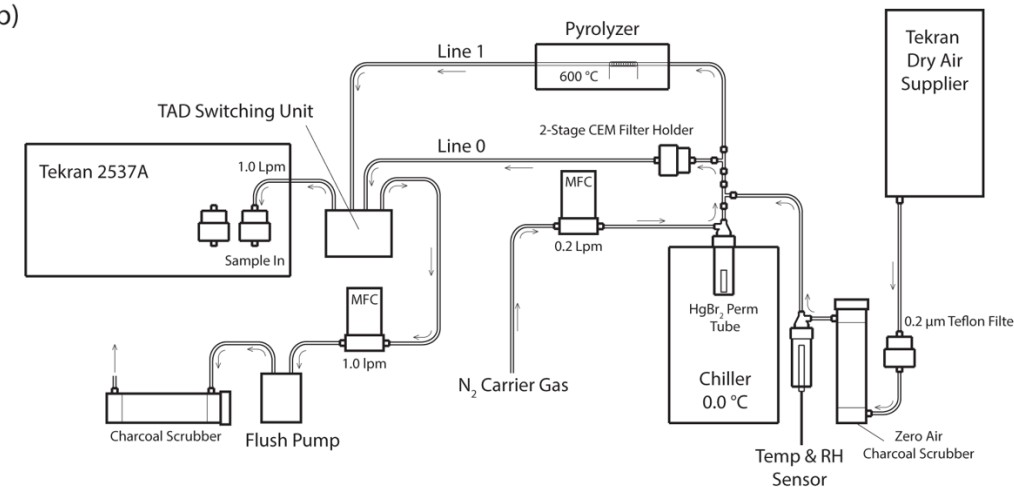

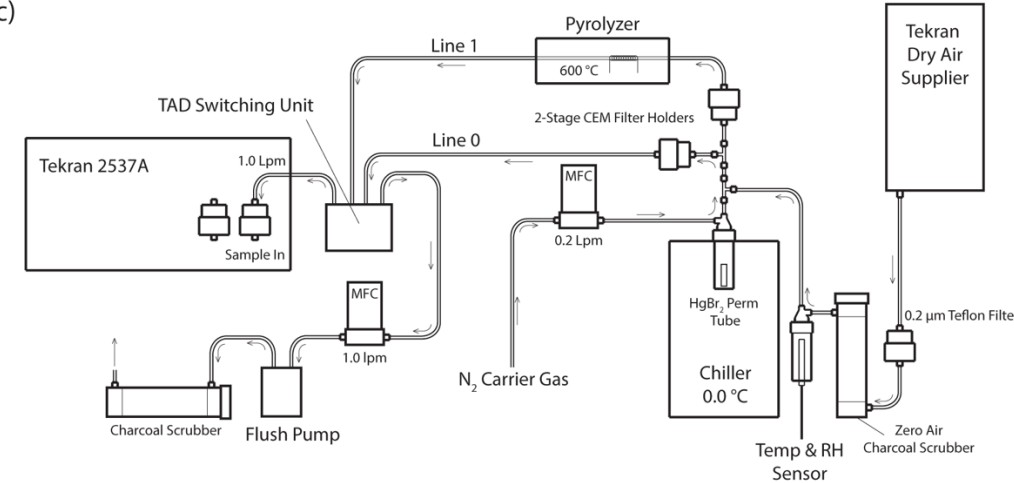


537        **Figure 1.** Schematic of the Hg vapor permeation system configurations for: a) GEM permeations b) $HgBr_2$
permeations c) Simultaneous $HgBr_2$ loading on two sample lines. Note dry air supplier disconnected for ambient and
elevated humidity $HgBr_2$ permeations, with sample path starting at 0.2 µm Teflon particulate filter and water bath
inserted immediately in front of the charcoal scrubber. All tubing is PTFE, except for the quartz glass pyrolyzer tube
541                                   and PFA filter holders.

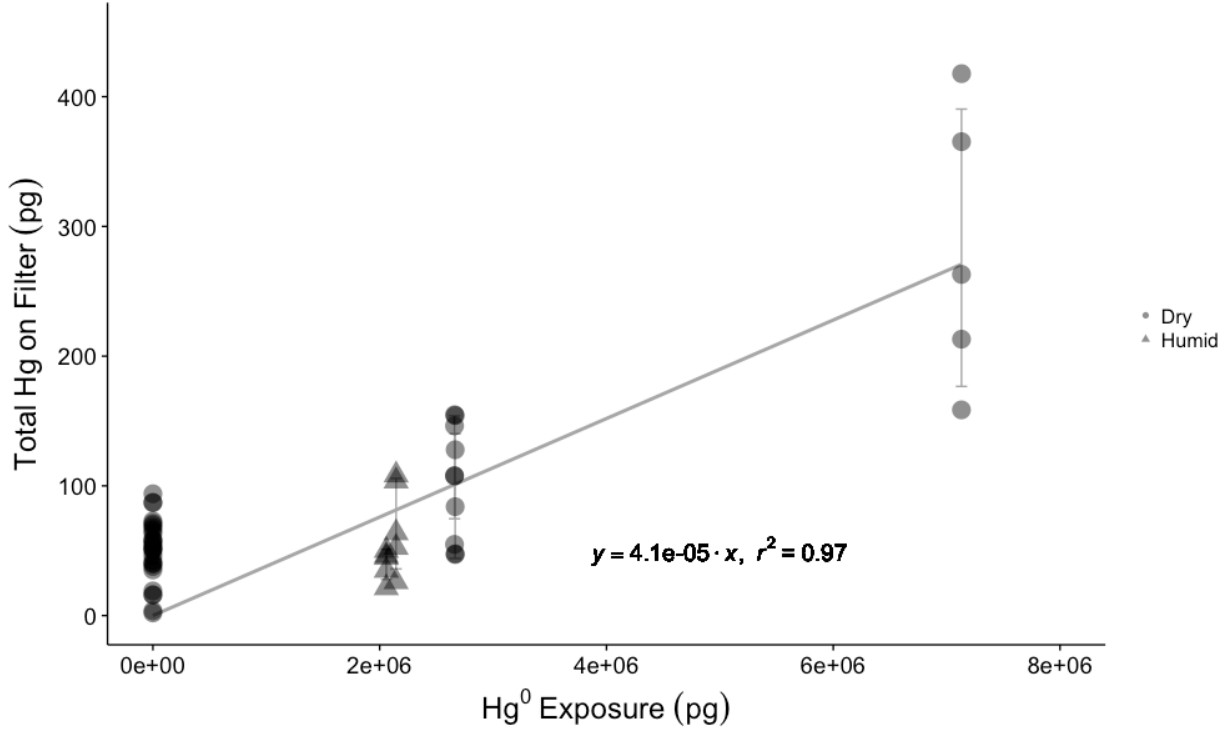


Figure 2. Total Hg recovered on CEM material for blank filters (Hg exposure = 0 pg) and different $Hg^0$ vapor permeations in dry ($0.5 \pm 0.1$ g m$^{-3}$ WV) and humid air (2-4 g m$^{-3}$ WV). Circles represent dry air permeations, triangles represent humid air exposures, and all permeation exposures were blank-corrected. The regression line shows the relationship between total $Hg^0$ exposure and blank-correct mean total Hg recovered on CEM filters (error bars $\pm$ one standard deviation), with a slope of $4.1 \times 10^{-5}$ indicating a linear uptake rate of 0.004%.



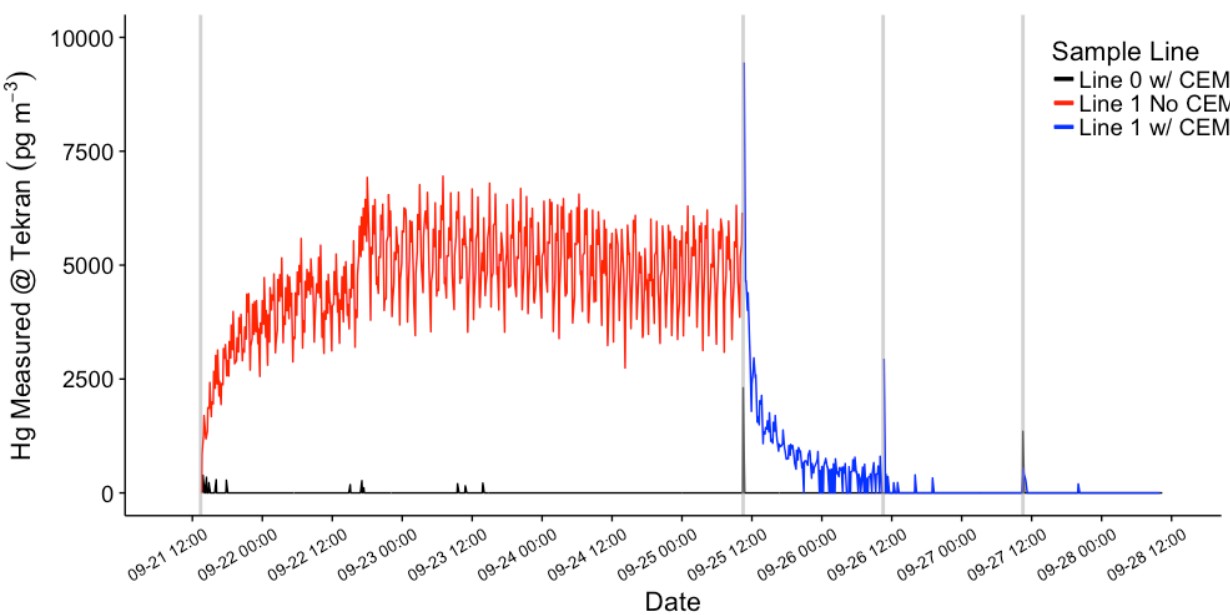

**Figure 3.** HgBr$_2$ permeations in clean dry lab air using the configuration in Figure 1B (red line) and Figure 1C (blue
line). The red line indicates total Hg released from permeation tube and passing through pyrolyzer on Line 1 before
being measured by Tekran 2537A, black line indicates Hg reaching 2537A through CEM filters on Line 0. Vertical
grey lines indicate open system during filter deployments.

**Table 1**.

| Sample | Start | End | Sample Time (min) | Sample Flow (lpm) | Sample Volume (m³) | Total Hg on CEM (pg) | Blank Correct (pg) | Total Hg @ Tekran (pg) | A to B Filter Brkthru (%) |
|---|---|---|---|---|---|---|---|---|---|
| **Mean CEM Filter Blank** | | | | | | | 54 | | |
| | | | | | *Clean Dry Air (0.3 ± 0.05 g m⁻³ wv)* | | | | |
| *HgBr 1P* | *9/21/17 13:25* | *9/25/17 10:25* | *5580* | *1.00* | *5.580* | *na* | *na* | *25181* | *na* |
| HgBr 1A | 9/21/17 13:25 | 9/25/17 10:25 | 5580 | 1.00 | 5.580 | 49478 | 49424 | 15 | 0.10 |
| HgBr 1B | | | | | | 101 | 47 | | |
| HgBr 2A | 9/25/17 10:30 | 9/26/17 10:30 | 1440 | 1.00 | 1.440 | 8901 | 8847 | 0 | 0.20 |
| HgBr 2B | | | | | | 71 | 17 | | |
| *HgBr 3A* | *9/25/17 10:30* | *9/26/17 10:30* | *1440* | *1.00* | *1.440* | *9125* | *9072* | *1155* | *0.36* |
| *HgBr 3B* | | | | | | *86* | *33* | | |
| HgBr 4A | 9/26/17 10:40 | 9/27/17 10:25 | 1425 | 1.00 | 1.425 | 8494 | 8440 | 0 | 0.28 |
| HgBr 4B | | | | | | 77 | 24 | | |
| *HgBr 5A* | *9/26/17 10:40* | *9/27/17 10:25* | *1425* | *1.00* | *1.425* | *8306* | *8253* | *10* | *0.36* |
| *HgBr 5B* | | | | | | *83* | *29* | | |
| HgBr 6A | 9/27/17 10:35 | 9/28/17 10:25 | 1430 | 1.00 | 1.430 | 8496 | 8442 | 0 | 0.22 |
| HgBr 6B | | | | | | 72 | 19 | | |
| *HgBr 7A* | *9/27/17 10:35* | *9/28/17 10:05* | *1410* | *1.00* | *1.410* | *8386* | *8333* | *6* | *0.15* |
| *HgBr 7B* | | | | | | *66* | *13* | | |
| | | | | | *Clean Humid Air (4.4 ± .2 g m⁻³ wv)* | | | | |
| *HgBr H1P* | *10/2/17 16:10* | *10/3/17 15:20* | *1390* | *1.00* | *1.390* | *na* | *na* | *5888* | *na* |
| HgBr H1A | 10/2/17 16:10 | 10/3/17 15:20 | 1390 | 1.00 | 1.390 | 10498 | 10444 | 1700 | 0.25 |
| HgBr H1B | | | | | | 80 | 27 | | |
| HgBr H2A | 10/3/17 15:30 | 10/4/17 14:40 | 1390 | 1.00 | 1.390 | 8589 | 8535 | 164 | 0.13 |
| HgBr H2B | | | | | | 65 | 11 | | |
| *HgBr H3A* | *10/3/17 15:30* | *10/4/17 14:40* | *1390* | *1.00* | *1.390* | *8182* | *8129* | *420* | *0.54* |
| *HgBr H3B* | | | | | | *98* | *44* | | |
| HgBr H4A | 10/4/17 14:50 | 10/5/17 11:50 | 1260 | 1.00 | 1.260 | 7504 | 7451 | 0 | 0.31 |
| HgBr H4B | | | | | | 76 | 23 | | |
| *HgBr H5A* | *10/4/17 14:50* | *10/5/17 11:50* | *1260* | *1.00* | *1.260* | *7576* | *7522* | *25* | *0.25* |
| *HgBr H5B* | | | | | | *73* | *19* | | |
| *HgBr H6P* | *10/5/17 12:05* | *10/9/17 10:25* | *5660* | *1.00* | *5.660* | *na* | *na* | *11889* | *na* |
| HgBr H7A | 10/9/17 10:40 | 10/10/17 10:45 | 1445 | 1.00 | 1.445 | 9024 | 8970 | 105 | na |
| HgBr H7B | | | | | | 2672* | 2618* | | |
| *HgBr H8A* | *10/9/17 10:40* | *10/10/17 10:45* | *1445* | *1.00* | *1.445* | *12359* | *12305* | *397* | *na* |
| *HgBr H8B* | | | | | | *75* | *21* | | |
| | | | | | *Clean High Humidity Air (10.9 ± 1.7 g m⁻³ wv)* | | | | |
| HgBr H9A | 10/10/17 10:50 | 10/11/17 9:30 | 1360 | 1.00 | 1.360 | 10920 | 10866 | 181 | 0.22 |
| HgBr H9B | | | | | | 78 | 24 | | |
| *HgBr H10A* | *10/10/17 10:50* | *10/11/17 9:30* | *1360* | *1.00* | *1.360* | *11413* | *11359* | *308* | *0.00* |
| *HgBr H10B* | | | | | | *53* | *0* | | |
| HgBr H11A | 10/11/17 9:35 | 10/12/17 9:35 | 1440 | 1.00 | 1.440 | 12001 | 11947 | 5 | 0.00 |
| HgBr H11B | | | | | | 52 | 0 | | |
| *HgBr H12A* | *10/11/17 9:35* | *10/12/17 9:35* | *1440* | *1.00* | *1.440* | *12579* | *12525* | *40* | *0.29* |
| *HgBr H12B* | | | | | | *90* | *36* | | |
| *HgBr H13P* | *10/12/17 9:40* | *10/13/17 9:40* | *1440* | *1.00* | *1.440* | *na* | *na* | *1430* | *na* |
| HgBr H13A | 10/12/17 9:40 | 10/13/17 9:40 | 1440 | 1.00 | 1.440 | 13152 | 13099 | 4 | 0.12 |
| HgBr H13B | | | | | | 69 | 16 | | |


**Table 1.** Summary of CEM filter loading and breakthrough during HgBr₂ permeations. Samples denoted P indicate
approximate permeation rate check through Line 1 via pyrolyzer and Tekran 2537A, italicized text indicates filter
deployments on Line 1, and * indicates high values due to leak around first filter seal.