# Peer review of "Evaluation of cation exchange membrane performance under exposure to high Hg0 and"

_Atmospheric Measurement Techniques, 2018_

## Referee Comment (RC1) · Anonymous Referee #2 · 4 Jul 2018

While the quality of writing within this manuscript is very good (a major reason for accepting the quick review), upon a more detailed analysis there are some fundamental concerns with the quality of the science within the manuscript. The experimental design appears rushed and incomplete, to the point that some methodological issues were even brought up by the authors themselves, which could have easily been addressed and experiments repeated were not done. As such I cannot accept it for publication in its current state.

Major issues that require addressing:

1.  No standard reference material used to confirm recovery of the acid digestion

method. I have a major concern about the lack of inclusion of reference materials for quality assurance and control purposes. Especially given the elevated recoveries of RM above what measured by the Tekran/pyrolyzer. Without any such assessment how can we rule out contamination from instruments, handling equipment or even uptake of outside Hg into the acid digestates? Blank filters could have easily been spiked with a sediment or soil SRM and analysed to confirm the recovery of the analytical method. Also, in response to this comment, it is not acceptable simply quote another paper that has done this. It needs to be confirmed in the lab and experimental settings used in these experiments.

2. Concerns with the pyrolyzer conditions used in analysis. There is no comment on the performance of the pyrolyzer not being 100% efficient in reducing $HgBr_2$ to $Hg0$ in the methods section (only in the SI where the majority of readers will not see such a concern); a major issue when discussing Hg recovery discrepancies on the filters and the Tekran analyser . If the pyrolyzer is not performing at 100% how can we be certain the system is collecting all the mercury? They go on to suggest higher pyrolyzer temperatures above 600 C would improve its performance. Indeed Lynam and Keeler (2002) suggest that pyrolyzer temperatures up to 900 C may be necessary. So why did the authors not repeat the experiments with higher pyrolyzer temperatures as they suggest? The fact that they did not implies incompleteness of the experiment. Results would be greatly improved with a more efficient pyrolyzer, and the exact nature of CEM recoveries vs Tekran/pyrolyzer recoveries may have been revealed.

3. Removal of 1st CEM traps from GEM breakthrough experiments. This I totally disagree with. The teflon lines could easily be cleaned by (a) rinsing the lines in an acid solution then DI water and allowing them to dry in zero or very low Hg air and (b) running zero Hg air through the system with CEMs in place before the actual GEM permeation cycles. At this point there should be no RM in the system. Thus for the GEM permeation runs why would you discard the first set of CEM filters without this analysis? There should be no RM in the system and any collected Hg should be

assumed to be GEM inadvertently collected. Again the discussion in the SI is HIGHLY relevant and ignored in the main body of the paper. The first set of CEMs were always higher and not arbitrarily so (as the authors seem to suggest), but this is not mentioned in the main paper only the SI. "We believe it is unlikely that the Hg observed on the first CEM filters results from GEM uptake." The Hg uptake on the first set of filters is attributed to residual GOM in the lines, but if lines were properly cleaned before analysis this would not be the case. This is something that could have been ruled out one way or another through subsequent analysis and again no doing so implies the experiments are incomplete. Furthermore if it the 1st CEMs were picking up "residual RM" from the lines then they would not have seen the dramatic increase in the CEM filter concentration under higher GEM concentrations. GOM was not produced under this scenario and therefore the "residual RM" should not increase, but as SI Figure 4 shows it did increase and exponentially, not linearly.

4. Teflon lines were not heated and line lengths not fully described. Higher recoveries on CEMs could be associated with losses in HgBr2 to the longer inlet line on line 1 as it appears in Figure 1(b). Description of the length of tubing between the switch valve and the pyrolyzer (line 1) and the switch value and the CEMs on line 0 should be included. Any difference in length in unheated lines may also be causing inconsistent recoveries. Heating the lines (common practice in atmospheric Hg monitoring to include GOM in analysis) would reduce any such losses. This is another simple adjustment that would have produced more complete experimentation.

5. Sorption of all forms of Hg to CEMs are assumed to be the same at low concentrations as they are in these high concentration experiments. Maybe this is the case, but it could have easily been proven by repeating experiments (for longer time periods) at much lower concentrations and I can't really see why such experiments would not be included once things were already set up.

6. High blank levels are a concern for background sampling. While blank levels of 50 $\pm$ 20 pg may not seem high given the very high concentrations used in these experiments

at background concentrations of RM this could be an issue. Method Detection limits MDL= 3*SD of blanks = 60 pg Method Quantification limits MQL= 10*SD of blanks = 100pg Assume background RM concentration of 10 pg/m3 or 0.01 pg/L Flow rate = 1 L/m This translates to 0.6 pg/hr being sorbed to the CEM, which would require 100 hours of sampling at background levels to exceed just the MDL and about 330 hours to reach MQL. These issues are not currently discussed adequately in the manuscript nor covered by the experimental design of the manuscript.

Other general concerns:

1. More caution should be used in the definition of the term reactive mercury (RM). While it no doubt has some use, combining GOM and PBM as RM is diluting specific information by grouping together two already very broad classes of atmospheric Hg species. Our lack of knowledge and understanding of the molecules and complexes that make up the specific forms of GOM and PBM is a major driving force behind differences between global Hg transport and fate models and measured values, our poor understanding of atmospheric Hg cycling and even terrestrial - atmospheric inter-actions. Using RM to describe both species does little to improve that understanding. Furthermore, while the use of the term does exist in the literature it is not widely applied beyond one or two research groups. The sampling method applied in this manuscript cannot distinguish between GOM and PBM, thus RM must be used here, but a much greater description of this caution must be given in the manuscript.

2. The use of HgBr2 as a surrogate for all "RM". As I have just mentioned we do not know nearly enough about what the exact species of GOM or PBM (let alone both combined as RM) are. Different species of GOM and PBM are likely to behave quite differently in the atmosphere and indeed on different sorption media. I have concerns that using only HgBr2 as a surrogate (do we even know if HgBr2 is a common at-mospheric GOM constituent? – It has been suggested that Br acts as the primary oxidant, but the very reactive HgBr (1+) product has a very short lifetime of less than a second before other more stable Hg2+ compounds are produced through oxidation

by other atmospheric species (Horowitz et al. 2017)) . The data from this manuscript appear to show that CEMs effectively sorb HgBr2, but how do we know they sorb ALL species? Again this emphasizes the concern of using RM - a more generic term - to define sorption of both GOM and PBM. We should be focusing on determining what specific species and complexes that make up GOM and PBM rather than be even less specific and defining everything as RM, based on only injection of HgBr2. Again better acknowledgement of this methodological short-coming is needed to proceed.

3. The review of the literature in the manuscript is skewed quite favourably to CEMs and quite negatively to existing methods. A more rounded approach would be less evident of bias towards the CEMs (see specific comments).

Specific Comments:

Line 12: "Reactive mercury (RM)..." should be described here as: "Reactive mercury (RM), the sum of both gaseous oxidised Hg and particulate bound Hg,..."

Line 29: "...high collection efficiency." Should be changed to: "...high collection efficiency of the target analyte."

Line 35-36: This is where a cautionary description of the use of the term RM should be included.

Line 41-42: These reviews do provide a good critique of the Tekran based speciation measurement techniques, but they do not tell the whole story and more literature needs to be discussed here. For example, Marusczak et al. 2017, describes how adding the zero flushes from GOM analysis to the actual GOM concentrations increases the derived concentrations to agree more closely with alternative measurement techniques and some modelled values. Such advancements as the latter with the previous system should also be discussed to ensure impartiality. Additionally Cheng and Zhang (2017) state: "Other measurements techniques such as mist chambers, nylon and cation exchange membranes [CEMs], and Detector for oxidized Hg, were capable of collecting

more GOM than KCl-denuders. However, similar to the Tekran instrument, these alternative methods are not immune to sampling artefacts caused by high water vapour and other gases and aerosols." A similar degree of impartiality would greatly benefit this review of the literature.

Line 78: The analytical information in parenthesis should be deleted. This is describing another paper in too much detail and not needed here.

Line 88: Again as per the comment on lines 41-42 this is ignoring advancements in other methods and caveats of the CEM methodology.

Line 93-95: This information is extremely important and I credit the authors for its inclusion.

Line 111: "...with a view to estimate the collection efficiency and..." this absolutely should state: "...with a view to estimate the collection efficiency OF THIS ANALYTE and..."

Line 121-122: "Each of these materials is known to be chemically inert, virtually non-porous, and to have a low coefficient of friction." This needs to be referenced.

Line 134: "...an activate charcoal scrubber..." Please provide details of this scrubber: elemental imgregnation (if any) and manufacturer. Different activated charcoal scrubbers perform differently with regards to atmospheric Hg sorption with halogen and sulphur impregnated charcoals performing better for Hg sorption (e.g. Vidic et al. 1998).

Line 152: Were flows measured downstream of both sampling lines to ensure pressure differences across the CEM filters and the pyrolyzer did not cause flow rate differences into the two lines? Any difference in flow may result in differing recoveries of the two measurements.

Line 201: These recovery values should be adjusted to include the first CEM filter.

[Figure]

Line 218: "4540 pg m-3" it should be noted here or in the methods that this concentration is around 50-1000x higher than typical GOM concentrations and that behaviour may be slightly different under lower concentrations.

Line 224-226: While this does mean the absolute uptake capacity of the CEMs for HgBr2 is very high, we must remember that this is likely to be a thermodynamic/equilibrium parameter more than just a kinetic one and the uptake capacity will be higher under elevated conditions than at ambient conditions. This should be noted.

Line 244-246: But again this is contamination for very high concentrations. If just a small amount of this continues to be emitted during ambient sampling then this will represent a very substantial contamination of HgBr2. A cautionary note should be made about sampling management; systems used for higher concentrations should be only used for higher concentrations and likewise systems for lower concentrations. This will prevent any contamination from systematic memory effects.

Line 263: What would cause this reaction? Normally reduction in the presence of water is driven by photochemistry. Are these lines exposed to solar radiation? Please reference this suggested mechanism.

Line 279: "...(2 orders of magnitude above background)..." This should be 3 orders of magnitude. Background is ~1.5 ng/m3 the concentration used here is ~1500 ng/m3.

Line 305-308: So what does the difference in recoveries mean? Conclusions are meant to summarise what was found, but here we are just getting a rehash of the numbers without any explanation. Not sure I see any value in this.

Line 317-320: Again, the authors state a problem with the pyrolyzer, so why wasn't it optimised and experiments repeated?

SI Lines 14-15: What are the reported recoveries of the internal injections compared to the external injections, please provide details (%) and number of checks (n)

SI Lines 60-62: This is purely speculation based on almost no evidence. Just as likely

is it may be coming off the lines or indeed a little GEM is sorbing either to the CEMs or to something else on the CEMs. None of these scenarios can be fully ruled out using the data presented in these experiments. This statement is too speculative and should be removed

References:

Cheng, I. and Zhang, L., Environ. Sci. Technol. 2017, 51, 855−862.

Horowitz, H. et al., Atmos. Chem. Phys., 2017, 17, 6353-6371.

Lynam, M. and Keeler, G., Anal. Bioanal. Chem. 2002, 374, 1009-1014.

Marusczak, N. et al., Environ. Sci. Technol. 2017, 51, 863−869.

Vidic, R., et al., J. Air & Waste Manage. Assoc. 48, 247-255, 1998

---

## Referee Comment (RC2) · Anonymous Referee #1 · 27 Jul 2018

Miller et al. report long-awaited QA/QC testing of polyethersulfone cation exchange membranes (CEMs) for atmospheric divalent reactive Hg (RM) quantification. RM, operationally defined as the sum of particle bound Hg and gaseous oxidized forms of Hg (such as HgBr2, studied here) is difficult to pre-concentrate and quantify. Conventional, denuder-based, techniques are increasingly criticized and CEMs are an interesting alternative. The study addresses two key aspects: RM retention (the purpose of the CEMs) and Hg(0) retention (an unwanted artefact). The experimental design is well described and smartly designed; generally this is challenging work because HgBr2 is reactive and labile. Despite a number of experimental improvements (suggested by Referee 1) that could be explored in future work, I find that the authors are to be com-

mended for their rigorous approach. I appreciate that both the main text and SI contain descriptions of problems and unexpected issues; this is very important for future R&D on CEMs. The MS is generally well written and referenced. The findings that no significant Hg(0) artifact occurs at high levels, and that the RM is retained >99% efficiently, is encouraging news to the atmospheric Hg community. It is the key step in further deploying CEMs for atmospheric Hg speciation research and monitoring. I strongly recommend the MS for publication in AMT.

minor comments: L40 Please provide references for the 'demonstrated interferences and artifacts' of current GOM and PBM monitoring techniques. L181. "...by digestion in an oxidizing solution....(EPA)". I know that readers can look up the EPA method, but it could be informative to add a few details on the digestion method, for ex. (1N HCl, 12h, 80oC).

---

## Author Comment (AC1) · 5 Aug 2018

Response to Major Issues:

1. In response to the concern that no standard reference material is used during the acid-digestion analysis:

Mercury standards were in fact analyzed repeatedly during every CEM filter analysis. In the described method, standard additions of known mercury concentration are analyzed after every batch of 10-12 samples, in what are known as Ongoing Precision and Recovery (OPR) samples. This includes a mercury standard analyzed immediately

following generation of the calibration curve, and at least one standard analyzed at the end of the entire analysis. Acceptable recovery for these OPR standards is  $\pm$  10%.

The mercury standards are prepared at the same time and with the same batch of reagents as used for the sample filter digestion and analysis (i.e. ultrapure water, bromine monochloride, stannous chloride, etc.). The calibration blanks and filter blanks rule out contamination of any of the analytical equipment. No analysis proceeds where contamination is detected or suspected.

Prior publications and the EPA Method 1631 are here referenced because it describes in detail the way that acid digestions for total mercury are typically accomplished, and which are followed in this manuscript.

We will address this concern in the text by including a more thorough description of the analytical procedure and providing the specific OPR data for the analyses performed during this study.

2. In response to concerns about pyrolyzer efficiency:

Referee #2 highlights Lynam and Keeler (2002), who suggest that 800 C pyrolysis was perhaps insufficient for determining total mercury from particulate mercury filters, based on lower total Hg recoveries from thermal desorption and pyrolysis versus an acid digestion method.

Firstly, the Lynam and Keeler study deals primarily with particulate mercury, for which pyrolysis temperatures are necessarily higher in order to insure decomposition of the larger masses involved compared to pure gas-phase samples. Secondly, Lynam and Keeler (2002) also suggest that sample matrix interferents may have played a more important role in lower total Hg recoveries during thermal desorption and pyrolysis, via passivation of the analytical gold traps or interference with detection.

Additionally, for determination of RGM Lynam and Keeler (2002) use only a Tekran 1130 denuder module without an attached 1135. Consequently, RGM was determined

AMTD
exclusively from the standard 500 C denuder desorption temperature, with no further 800 C pyrolysis in the 1135 module. Lynam and Keeler are not alone in this practice, as many studies since have reported RGM/GOM values using only the Tekran 1130 denuder module without the benefit of further pyrolysis in the 1135 module (e.g. Temme et al., 2003; Weiss-Penzias et al., 2003; Soerensen et al., 2010; Wright et al., 2014; Huang and Gustin, 2015). The fact that the 1130 denuder module can even be operated without the inclusion of the 1135 and it's pyrolyzer is an implicit declaration by an industry-leading mercury analytics company that 500 C is adequate to desorb and decompose RGM/GOM.

Numerous publications have used lab-based pyrolyzer temperatures of 500-650 C when working with GOM compounds (e.g. Swartzendruber et al., 2009; Lyman et al., 2010; Lyman and Jaffe, 2011; Huang et al., 2013). The 600 C pyrolyzer temperature used in this study was a compromise based on what could realistically be achieved within the scope of this project and was within a well-established range of pyrolyzer temperatures used in previous work. The fact that the 600 C temperature appears to have been insufficient is an important result in and of itself and supports the need for higher pyrolyzer temperatures in future work.

The authors acknowledge the limitations of the pyrolyzer used in this study, point to the fact that these limitations are discussed in the text, and re-iterate that the main conclusions of this manuscript do not hinge on pyrolyzer performance. In fact, the majority of experiments described in this manuscript could have been successfully conducted with no pyrolyzer at all. Specifically, the generation of GEM was not contingent on pyrolyzer performance. Permeated GEM was generated using a new bead of analytically certified pure elemental mercury in a clean nitorgen flow. The pyrolyzer was used only as a backup to attempt to insure that all mercury was indeed GEM before reaching the CEM filters. HgBr2 loading during the simultaneous CEM filter deployments on Line 1 and Line 0 (per Figure 1C) was performed upstream of the pyrolyzer, and consequently these results are also entirely independent of pyrolyzer performance.
We will address concerns about pyrolyzer efficiency in the manuscript by discussing pyrolyzer needs and limitations at great depth.

3) In response to concerns over removal of first in-series CEM filter during GEM permeations:

The authors disagree with the contention that the first CEM filter in a series of 6 could somehow be sorbing GEM differently than the following 5 filters exposed to the same carrier flow and GEM concentration. Given the aforementioned uncertainties about absolute pyrolyzer efficiency, it seems much likelier that a small fraction of oxidized mercury is passing through the system and being scrubbed by the first CEM.

In regard to concerns about undemonstrated GEM sorption at low ambient concentrations, the authors are unfamiliar with any isothermal sorption mechanism whereby lower concentrations of an analyte result in higher rates of sorption compared to higher concentrations. If such a mechanism exists, we would appreciate guidance to the relevant literature.

However, any debate on this point is purely academic. Even when including the first CEM the observed GEM uptake would still be negligible (0.006% overall, 0.017% first filters only). If necessary, the authors see no difficulty including the first CEM in the analysis of GEM uptake, and this discussion can be moved into the text rather than the SI.

4. In response to the concern over unheated Teflon sample tubing length between Line 1 and Line 0:

For the configuration shown in Figure 1B, the authors acknowledge a difference in unheated Teflon tubing length between the filter pack on Line 0 and the pyrolyzer on Line 1.

However, the authors maintain that the more important results are obtained from the configuration shown in Figure 1C, in which the length of Teflon tubing from the per-
meation vial to each of the filter assemblies is exactly the same. In this configuration potential line-loss would be equivalent. Indeed, this is shown by the consistent HgBr2 loading on the CEM filters on Line 1 and Line 0.

5. In response to a desire for additional experimentation:

The authors agree that more experimentation is both welcome and necessary, and indeed is now ongoing within the scope of an expanded and fully funded NSF research project.

However, our original and primary concern within the more narrow scope of this manuscript was with GEM uptake and HgBr2 breakthrough at high loading rates. These questions were of particular importance to a companion manuscript, also currently submitted to AMT.

6. In response to concerns about high CEM filter blanks:

It is unclear to the authors how this constitutes a 'major issue' with the work presented in this manuscript.

The authors acknowledge that long sample periods are currently required when using the CEM filters in ambient air sampling. We did not specifically discuss ambient air sampling time requirements because this is not an ambient air sampling study, and the CEM filters are not used exclusively for ambient sampling. Previous work, which is referenced in this manuscript, clearly indicates that the CEM filters are deployed for 2-week (336 hour) sample periods in ambient air.

Putting aside the non-relevance of this concern in regard to the present manuscript, the authors would like to point out that filter blanks reported in this study are in fact lower than in previously published studies using the CEM methodology, and so constitute an improvement in methodology that should alleviate concerns about detection limits rather than exacerbate them.

Response to Other General Concerns:

AMTD
1. In response to the concern that more caution should be used in the definition of the term reactive mercury (RM):

This point is well taken, and the authors will elaborate on the limitations of using the term RM. However, the authors also point out that no current measurement technique can always reliably separate and quantify GOM and PBM, and there is uncertainty as to whether even total atmospheric Hg is always accurately measured. We suggest that the ability to accurately quantify total reactive mercury is a necessary first requirement to differentiating between GOM and PBM.

2. In response the concern over the use of HgBr2 as a surrogate for all "RM":

The authors acknowledge that the use of HgBr2 as a lone surrogate for GOM is a limitation of this manuscript. However, the results of the permeation tests using HgBr2 are positive, and therefore justify ongoing and intensive validation using a host of GOM compounds, under a greater range of conditions. Unfortunately, such expanded experimentation was beyond the scope of this study, but is now the objective of an ongoing fully funded project. If the limited permeation tests described in this manuscript had failed to demonstrate adequate performance of the CEM material, the authors would have duly presented such results and discontinued further use.

The results presented in this manuscript also invite and encourage replication by other researchers and other labs.

3. In response to concerns that the review of the literature in the manuscript is skewed quite favourably to CEMs:

We will certainly round out the literature review to be more even-handed.

Response to Specific Comments:

Line 12: "Reactive mercury (RM). . ." should be described here as: "Reactive mercury (RM), the sum of both gaseous oxidised Hg and particulate bound Hg,..."

AMTD
**Correction made**

Line 29: ". . .high collection efficiency." Should be changed to: ". . .high collection efficiency of the target analyte."

**Correction made**

Line 35-36: This is where a cautionary description of the use of the term RM should be included.

**Correction made**

Line 41-42: These reviews do provide a good critique of the Tekran based speciation measurement techniques, but they do not tell the whole story and more literature needs to be discussed here. For example, Marusczak et al. 2017, describes how adding the zero flushes from GOM analysis to the actual GOM concentrations increases the derived concentrations to agree more closely with alternative measurement techniques and some modelled values. Such advancements as the latter with the previous system should also be discussed to ensure impartiality. Additionally Cheng and Zhang (2017) state: "Other measurements techniques such as mist chambers, nylon and cation exchange membranes [CEMs], and Detector for oxidized Hg, were capable of collecting more GOM than KCI-denuders. However, similar to the Tekran instrument, these alternative methods are not immune to sampling artefacts caused by high water vapour and other gases and aerosols." A similar degree of impartiality would greatly benefit this review of the literature.

Will revise the literature review with greater impartiality

Line 78: The analytical information in parenthesis should be deleted. This is describing another paper in too much detail and not needed here.

Correction made

Line 88: Again as per the comment on lines 41-42 this is ignoring advancements in
other methods and caveats of the CEM methodology.

**Will revise**

Line 93-95: This information is extremely important and I credit the authors for its inclusion.

**Thank you**

Line 111: ". . .with a view to estimate the collection efficiency and. . ." this absolutely should state: "...with a view to estimate the collection efficiency OF THIS ANALYTE and..."

**Correction made**

Line 121-122: "Each of these materials is known to be chemically inert, virtually nonporous, and to have a low coefficient of friction." This needs to be referenced.

**Reference included**

Line 134: ". . .an activate charcoal scrubber. . ." Please provide details of this scrubber: elemental imgregnation (if any) and manufacturer. Different activated charcoal scrubbers perform differently with regards to atmospheric Hg sorption with halogen and sulphur impregnated charcoals performing better for Hg sorption (e.g. Vidic et al. 1998).

**Will provide details**

Line 152: Were flows measured downstream of both sampling lines to ensure pressure differences across the CEM filters and the pyrolyzer did not cause flow rate differences into the two lines? Any difference in flow may result in differing recoveries of the two measurements.

Both flows were controlled by MFC

Line 201: These recovery values should be adjusted to include the first CEM filter.
Can include if necessary

Line 218: "4540 pg m-3" it should be noted here or in the methods that this concentration is around 50-1000x higher than typical GOM concentrations and that behaviour may be slightly different under lower concentrations.

**Noted**

Line 224-226: While this does mean the absolute uptake capacity of the CEMs for HgBr2 is very high, we must remember that this is likely to be a thermody-namic/equilibrium parameter more than just a kinetic one and the uptake capacity will be higher under elevated conditions than at ambient conditions. This should be noted.

**Noted**

Line 244-246: But again this is contamination for very high concentrations. If just a small amount of this continues to be emitted during ambient sampling then this will represent a very substantial contamination of HgBr2. A cautionary note should be made about sampling management; systems used for higher concentrations should be only used for higher concentrations and likewise systems for lower concentrations. This will prevent any contamination from systematic memory effects.

**Cautionary note added**

Line 263: What would cause this reaction? Normally reduction in the presence of water is driven by photochemistry. Are these lines exposed to solar radiation? Please reference this suggested mechanism.

No solar radiation, but fluorescent room lighting during working hours. Anecdotally, more Hg observed downstream of the filters on the Tekran 2537 when lights turned on.

Line 279: ". . .(2 orders of magnitude above background). . ." This should be 3 orders of magnitude. Background is  $\hat{a}$ Lij1.5 ng/m3 the concentration used here is  $\hat{a}$ Lij1500 ng/m3.
**Correction made**

Line 305-308: So what does the difference in recoveries mean? Conclusions are meant to summarise what was found, but here we are just getting a rehash of the numbers without any explanation. Not sure I see any value in this.

Will revise conclusions

Line 317-320: Again, the authors state a problem with the pyrolyzer, so why wasn't it optimised and experiments repeated?

Because the main experiments described in this manuscript did not hinge on pyrolyzer performance SI Lines 14-15: What are the reported recoveries of the internal injections compared to the external injections, please provide details (%) and number of checks (n)

Details provided

SI Lines 60-62: This is purely speculation based on almost no evidence. Just as likely is it may be coming off the lines or indeed a little GEM is sorbing either to the CEMs or to something else on the CEMs. None of these scenarios can be fully ruled out using the data presented in these experiments. This statement is too speculative and should be removed

Statement removed

AMTD

---

## Author Comment (AC2) · 20 Aug 2018

The authors thank Referee #1 for their positive assessment.

Response to Minor comments:

L40 Please provide references for the 'demonstrated interferences and artifacts' of current GOM and PBM monitoring techniques.

We will revise to include references for this statement.

L181. ". . .by digestion in an oxidizing solution. . ..(EPA)". I know that readers can look up the EPA method, but it could be informative to add a few details on the digestion

method, for ex. (1N HCl, 12h, 80oC).

We will elaborate on the methodology, and as per Referee #2 we will include method calibration data as well.

---

## Author Response (AR2)

8 December 2018
Dear Dr. Abbatt,
Thank you for the opportunity to revise our manuscript for further consideration by Atmospheric
Measurement Techniques. The suggestions made by the reviewers were extremely helpful for
reworking our manuscript in a manner that, we hope you will agree, has made it a much stronger
contribution to the literature. We seriously considered each comment, and in this document we
attempt to address each comment and suggestion. To address these comments we added
additional data as well as moved material in the Supplemental Information to the main paper. In
addition, we changed the conclusions to a discussion that seems more appropriate and added a
brief section describing future work that is needed Below are the original reviewer comments
and suggestions in italics and our responses. Accompanying this letter are three documents: 1)
manuscript with edits tracked for easier review, 2) manuscript with all edits incorporated and not
tracked, and 3) updated supporting information. Again, thank you for the opportunity to submit
our revised manuscript to AMT and we look forward to further feedback.
Sincerely,
Sarrah M. Dunham-Cheatham, Postdoctoral Scholar
Mae Sexauer Gustin, Professor
Matthieu B. Miller
**Detailed Response to Reviewers:**
**Reviewer 1:**
*Evaluation of CEMs for GEM uptake and GOM capture and retention for use in a flow-based*
*technique is an important step towards obtaining realistic measurements of reactive mercury.*
*This study presents data that shows promising results for laboratory air. Of course, further work*
*using an ambient air matrix and other species of GOM is need.*
*I am concerned there was no mixing volume downstream of the bromide permeation tube. In*
*trying to keep line lengths short to minimize wall losses there may not be adequate mixing to*
*ensure equal concentrations travel through Line 0 and Line 1. This may bias uptake and capture*
*measurements and may have contributed to the mass balance problems. It may be one of the*
*causes of the large amount of noise in the pyrolyzed signal shown in Figure 3. If mixing volume*
*use was not possible then only running with a single Line should have been considered.*
We performed a control experiment to ensure that both Lines 0 and 1 were conducting
comparable concentrations of mercury under the experimental conditions. We deployed 2-stage
filter packs with cation exchange membranes (CEM) in each line at equal distances from the
permeation tube. The membranes were deployed for the same amount of time, in triplicate
deployments, and analyzed to quantify the amount of mercury sorbed to the membranes. The
average % deviation between lines was 2.9%, with a maximum deviation of 5.4%. These results
indicate to us that, though there may be some difference in the amount of mercury passing
through Lines 0 and 1, the difference is relatively small. These results have been added to the
manuscript to help alleviate the concern from readers.

*I am also concerned about the poor performance of the pyrolyzer. Inadequate conversion due to*
*residence time and temperature problems is fairly easy to address. It would have been a much*
*better paper if simple tests of increasing the pyrolyzer temperature had been done. Incomplete*
*reduction or varying efficiency may be another cause of the noisey signal in Figure 3.*

Additional experiments were conducted to ensure that the temperature of the pyrolyzer was
sufficient to convert GOM to GEM. Our results show that, under our experimental conditions,
mercury concentrations measured by the downstream Tekran 2537 were not significantly
different when the pyrolyzer temperature was 650 C and 800 C. These results have been added to
the manuscript supporting information. We agree that the residence time in the pyrolyzer design
is short, and that improvements to the pyrolyzer are needed. These are foci of ongoing studies.

*Another issue that was not raised in the paper is the effect of back-reactions downstream of the*
*pyrolyzer to reform HgBr2. What is the fate of the bromine?*

These are great questions and certainly warrant investigation. However, answering this question
is beyond the scope of the current project and assumptions would be unsupported by the
presented data; thus, we have decided to leave this discussion out of the manuscript.

*Other Comments:*

*Line 43: Not ppt but parts per quadrillion*

Thank you to the reviewer for catching this error. The correction has been made.

*Line 145: Locating the AC scrubber downstream of the humidifier could cause problems: the*
*carbon's ability to uptake GEM is reduced at higher humidity. Some AC scrubbers contain*
*iodine which can be released and migrate with humidity, which can then trap Hg downstream.*

While we recognize this as a potential issue, the scrubber was located downstream of the
humidity source to remove any potential contaminant Hg coming from the DIW used in the
humidity source. Our results indicate that at absolute humidity concentrations around 5 g m$^{-3}$, the
water vapor did not have a significant effect on the permeation system performance. However, in
the elevated humidity experiment, it was clear that the water vapor resulted in a large pulse of
mercury moving through both Lines, and discuss these results in the paper.

*Line 158: Was the dry N2 flow included when calculating the humidity values?*

No

*Line 165: The volume of pyrolyzer seems small. What is residence time? Were any tests done to*
*ensure that reduction was quantitative?*

The residence time in the pyrolyzer tube is approximately 1.5 seconds. Quartz wool was added to
the pyrolyzer to increase the amount of surface area available to facilitate reactions and
maximize the amount of GOM converted to GEM in the pyrolyzer. We have added this to the
text and commented on the fact this is a pretty efficient method for converting GOM to GEM. In addition a comment was added that indicated that having an efficient pyrolyzer allows us to
better constrain permeation rates.
*Line 327: How did you estimate line losses?*
The estimate of line loss was and is described on lines 321-325 of the manuscript.
*Line 330: I can believe that the pyrolyzer was not 100% efficient, but did you try to increase the*
*temperature and look for increased conversion?*
Yes, we performed an experiment to test if more mercury was converted to GEM, as indicated by
an increased measured mercury concentration in the downstream Tekran 2537, with increased
pyrolyzer temperatures. There was no significant increase in measured mercury concentrations
when the pyrolyzer temperature was increased from 650 C (circa the experimental pyrolyzer
temperature) to 800 C. When the pyrolyzer was increased to 1,000 C, significantly more mercury
was measured; however, the pyrolyzer design could not be safely sustained at an operational
temperature of 1,000 C, and thus due to safety concerns the pyrolyzer operating temperature was
set to 600 C. Pyrolyzer improvements in this permeation system are ongoing.
*Figure 1: It would be nice to show humidifier in this figure. I am concerned about the lack of*
*mixing volume downstream of the permeation gas TEE. Without adequate mixing there may not*
*be equal concentration of HgBr2 going to Line 0 and Line 1. This may be one of the reasons that*
*there is so much noise of the red trace of Figure 3 and could throw off your mass balance*
*calculations.*
The humidifier location was included in the figure caption and we prefer to leave it there rather
than add it to this very busy figure.
The mixing volume downstream of the permeation tube is small, by design. We tested for
evenness in mercury concentrations reaching each line (as described in more detail above), and
found that the average deviation in mercury concentrations between Lines 0 and 1 was 2.9%.
*SI Figure 1: The gaps between 2537 calibrations is about a month. Why wasn't it done on a*
*regular basis or at the end of each test?*
The system was not operated with an automatic internal calibration cycle to avoid interrupting
data collection during experiments. Due to the variable timing and duration of experiments,
calibrations were performed between experiments, which in some cases resulted in calibrations
that were several weeks apart. However, the calibration data presented in Figure S1 indicates that
the system was stable for the duration of the experiment.
**Reviewer 2:**
*Response to Major concerns:*
*1. In response to the concern that no standard reference material is used during the*
*acid-digestion analysis:*
*R2 – The reviewers have adequately addressed this concern*
*2. In response to concerns about pyrolyzer efficiency:*

*R2 – The authors response is inadequate in regard to the pyrolyzer. The authors attest to the*
*pyrolyzer working at 500 C for GOM by referencing other papers, but their own data shows that*
*there is a rather large discrepancy between the Tekran concentrations and the GOM*
*concentration in this dataset of which the authors acknowledge. This could be (1) inefficient*
*reduction of all GOM to GEM for subsequent Tekran analysis or (2) some other artefact in the*
*experiments. Using the Tekran was a clever way to QA/QC check the concentrations being*
*sorbed to the CEMs. But there was substantial discrepancies in the recoveries. To simply say*
*"the pyrolyzer was inefficient" and then argue in response to my criticism of this by showing*
*references stating that the pyrolyzer should work under at the tested temperatures is inadequate*
*when clearly something was awry in the experiments in terms of these recoveries. The authors*
*could have and should have EASILY re-tested the experiments at the pyrolyzer temperatures they*
*themselves suggest. It is my opinion that this simple, additional experimental treatment needs to*
*be included in the study. This relates to point 5 below.*
We performed a test of the pyrolyzer to test the conversion efficiency at increasing temperatures.
We have provided more details on this test in the comments above to Reviewer 1. In short, we
found that increasing the pyrolyzer temperature to 800 C did not result in a significant increase in
conversion of mercury to GEM. We have added the data from this experiment to the manuscript.
*3. In response to concerns over removal of first in-series CEM filter during GEM permeations:*
*R2 - Here I feel the authors have adequately addressed the concern by listing both the*
*proportion of sorbed GEM with and without inclusion of the first filter. It is still my preference*
*that the first filters be included and, as the authors state, this is still a very small amount. This*
*would also reduce the length of the manuscript and discussion on this could easily be moved to*
*the SI without affecting anything, which would help with the suggestion I make in point 5 below.*
*Nonetheless, one concern that is still yet to be addressed is why is the uptake of GEM on the first*
*filter not linear with increasing GEM concentration? It is exponential. I brought up this concern*
*in the previous round of comments and it was ignored.*
There are only five data points, so we feel any assertion about this relationship being linear or
exponential is not robust. That being said, the relationship is strongly linear ($r2 = 0.98$, p =
0.001).
*4. In response to the concern over unheated Teflon sample tubing length between Line*
*1 and Line 0:*
*R2 - Indeed another factor that may be contributing to the poor recovery results and another*
*part of the experiments that could have easily been adjusted and retested. Possibly not a big*
*issue as the authors suggest, but I imagine it also could have been adjusted easily.*
We certainly cannot rule out the possibility that the room temperature lines resulted in line loss
and contributed to the decreased recovery. However, we estimate our line loss to be a small
contribution to the decreased recovery.
*5. "The authors agree that more experimentation is both welcome and necessary, and*
*indeed is now ongoing within the scope of an expanded and fully funded NSF research*
*project. However, our original and primary concern within the more narrow scope of this*
*manuscript was with GEM uptake and HgBr2 breakthrough at high loading rates.*
*These questions were of particular importance to a companion manuscript, also currently*

*submitted to AMT."*
*R2 – What the authors are attempting to do here is manuscript quantity over manuscript quality.*
*These experiments show useful data, but as I have suggested they appear rushed and too many*
*questions and concerns have been left unanswered and unaddressed, when they could have*
*easily been rectified by simply re-running experiments making slight adjustments to the existing*
*conditions. This is THE STUDY addressing "GEM uptake and HgBr2 breakthrough at high*
*loading rates" it should not need "companion" studies or "follow-up" studies to*
*comprehensively answer the research question it set out to do. That is the role of this manuscript.*
*It is my suggestion that this study be combined with this "companion study" to make a more*
*complete paper. There is a lot of over-elaboration in the discussion and many points could easily*
*be moved to an SI without loss of information to combine this study with complimentary work.*
*This study definitely has merit, but it needs to be more polished.*
We appreciate the comment and have made every attempt to turn this manuscript into a more
polished version. We selected not to combine the work in this manuscript with the work of the
companion manuscript to avoid an extremely long, cumbersome manuscript that addresses two
different, yet complementary, sets of questions. We posit that the two manuscripts are stand-
alone studies and respectfully decline to combine them.
*6. In response to concerns about high CEM filter blanks:*
*R2 – This concern was brought up here because it has been inadequately addressed in previous*
*studies. Take the Huang et al (2017) study for example:*
*"The bi-weekly MDLs (336 h) for active systems with cation-exchange and nylon membranes*
*were 2–68 pg m−3 (mean: 24 pg m−3) and 0.01–14.6 pg m−3 (mean: 2.1 pg m−3), respectively.*
*Biweekly MDL was calculated from 3 times the standard deviation of bi-weekly blanks. The*
*MDL was calculated for each period of sampling, due to the fact this can vary based on*
*treatment of the membranes, the time samples are prepared for deployment, deployment at the*
*field site, and handling once returned to the laboratory. The membranes may also vary by*
*material lot. All samples were corrected by subtracting the blank for the corresponding 2-week*
*period."*
*(a) A 2 week mean detection limit concentration of 24 pg m-3 is at or below typical GOM*
*concentration even in industrial city sites (examples: Lyman and Gustin 2009; Huang et al.*
*2012; Choi et al. 2013).*
The Huang et al. (2017) reference uses the UNRRMAS system and is not comparable to the
system in this study. However, the high blanks (above background) can be attributed to the fact
that the CEM material passively collects (and accumulates) ambient GOM. Compared to the
actively sampled membranes, the "blank" membrane mercury concentrations are low and are
used to represent the amount of Hg sorbed to the material due to variables during preparation,
deployment, harvesting, processing, etc.. Thus, "blank correcting" the samples is appropriate in
these (actively sampled and/or high concentration) systems. See part (c) below for more details.
*(b) These are DETECTION limits. Not QUANTIFICATION limits. Attempting to provide high*
*certainty, quantifiable results at concentrations at or below detection limits is simply erroneous.*

We agree that reported concentrations should be above MDLs, and that a quantification limit
should be established for each instrument and method in each laboratory. We acknowledge that
these are of often 2 different values, with quantification limits typically higher than MDLs. In the
current study, we have only reported values above the quantification limits for the respective
instruments, and any value below the quantification limit is reported as ND or 0, where
appropriate. Additionally direct comparison of the 2 studies is not possible since for one the
blank is per unit of digestate, whereas the other one is Hg per volume of air.
*(c) The purpose of averaging method detection and/or quantification limits is to prevent exactly*
*what the authors are discussing here by smoothing out inconsistency in variable handling of*
*materials, residual Hg, etc. By subtracting "the blank" (a single blank for a single sampling*
*period; n=1) is again incorrect and biased.*
Presumably the reviewer is addressing the sentences in lines 203-206 of the original submission.
A total of 50 blank membranes collected over the course of this study were used to represent an
average background mercury concentration that could be attributed to the unused CEM material
due to manufacturing, shipping/transport, processing (cutting 47 mm rounds from the sheeted
material), material storage, sample deployment, sample harvesting, and sample analysis. The
blank membranes were handled in the same manner as the sample membranes, and thus
represent the cumulative contribution of mercury to the lot of membranes from the numerous
potential sources listed above. By subtracting the average blank mercury concentration from the
concentration of each sample membrane, we are quantifying the mercury concentration on the
sample membrane that resulted from experiment itself. This, of course, is not the ideal method
for determining the background concentration of each sample membrane; however, because the
analytical procedures are destructive, it is impossible to determine the background mercury
concentration on each membrane prior to deployment and therefore we have selected to use the
method discussed above and in the manuscript.
*(d) This completely contradicts a statement the authors of the current manuscript have*
*introduced in this round of reviews in terms of their attempting to caution the use of the term*
*RM:*
*"a broad term that favors basic accuracy of measurement over determination of specific*
*compounds"*
*How can this favor "accuracy of measurement" if attempts at field measurements at typical*
*concentrations encountered in the environment are at or below detection (not quantification)*
*limits. There is no confidence in such results. As such these concerns do still need to be*
*addressed for this sampling method.*
The phrase copied above has been removed from the manuscript.
*Other comments of this latest version:*
*Lines 39-40: This "cautionary" description of RM that was suggested by me previously is*
*inadequate as described above. This research group attempts to use this same sorbent material*
*to determine specific compounds of GOM in previous work (Huang et al. 2017). Thus, by making*
*this statement here they are saying the results from this previous work are not accurate*
*measurements.*

The second half of this sentence was removed. It added confusion to the discussion.

*Line 95: Once again here the authors should include a reference to the Marusczak et al. 2017*
*that describes how adding the zero flushes from GOM analysis to the actual GOM*
*concentrations increases the derived concentrations to agree more closely with alternative*
*measurement techniques and some modelled values. This was paper was specifically mentioned*
*in the previous round of comments to be added to the literature review to balance impartiality*
*but this comment was ignored.*

The reference was in fact added during the last revision We have added an additional citation to
this reference in this revision.

*Lines 128-131: Can the authors please reference where exactly in the Gustin et al. 2015 paper is*
*the mention of PTFE/PFA producing zero sorption of GEM. Personally, I could not find this*
*specific point within the reference.*

This sentence has been removed.

*Lines 220-222: Yes, the uptake was linear if the first filter was not included. But when the first*
*filter is included it is exponential. Why? As yet this has not been addressed by the authors*
*anywhere.*

We believe that this has been addressed, in response to an above comment and in the manuscript.
The first filters clearly do not group with the other samples, despite exposure to essentially
identical GEM concentrations We will not assign the behavior of the first filters to GEM uptake
because we cannot be sure they are not capturing some small amount of GOM, despite our best
efforts. However, we can be very sure that the subsequent downstream CEM filters are being
exposed only to GEM, given that the first filter would scrub over 99% of fugitive GOM.
Therefore, we feel the downstream filters to be the best representation of GEM uptake behavior
on the CEM material.

*Line 232: Approximate not approximately.*

This correction has been made.

*Lines 236-237: "but it should be noted that the performance of the CEM filters at low*
*concentrations could be slightly different"*
*This should be changed to:*
*"but it should be noted that these concentrations are 50-1000x above typical background*
*concentrations and the performance of the CEM filters at low concentrations could be slightly*
*different"as was previously suggested.*

We apologize for the oversight. This statement has been adjusted in the manuscript to reflect the
reviewer's suggestion.

*Lines 263-266: Caution needed here. You would never use such a system that has been used in*
*contaminated environments for background work. Even a small memory of the Hg would*
*overwhelm the background signal. Lines would need to be discarded or thoroughly acid cleaned*

*for background work. Please make a cautionary note on this here. A comment was made about this in the previous revisions, but ignored.*

This comment was not ignored. We have very clearly indicated that this is a laboratory system used exclusively for high concentration permeation work and is never used for background measurements. However, we have added additional elaboration on this point to the manuscript.

*Lines 284-285: Again I ask is "photochemistry driven by room fluorescent lighting" a process reported in the literature? If so please reference, if not this is just pure speculation.*

This statement has been adjusted to remove speculation.

*Lines 340-343: Again, why speculate when this could have easily been determined by repeating the experiments at higher pyrolyzer temperatures?*

As mentioned above, the pyrolyzer temperature did not significantly affect the amount of mercury transformed by the pyrolyzer to GEM when the temperature was increased to 800 C. Though higher pyrolyzer temperatures (1,000 C) transformed significantly more mercury, the temperature was unsustainable and unsafe for the pyrolyzer design. Therefore, the lower (safer) pyrolyzer temperature was used with the tradeoff being that not all GOM was converted to GEM and detected.

**Revised Manuscript, Tracked Changes:**

[revised manuscript text omitted]

---

## Author Response (AR3)

January 2019

Dear Dr. Abbatt,

Thank you again for the opportunity to revise our manuscript for further consideration by Atmospheric Measurement Techniques. The suggestions made by the reviewer were fair and have further improved our manuscript. Below are the original reviewer comments and suggestions (in italics) and our responses. Accompanying this letter are the: 1) manuscript with edits tracked for easier review, 2) manuscript with all edits incorporated and not tracked, and 3) revised supporting information. Again, thank you for the opportunity to submit our revised manuscript to AMT and we look forward to further feedback.

Sincerely,

Sarrah M. Dunham-Cheatham, Postdoctoral Scholar
Mae Sexauer Gustin, Professor
Matthieu B. Miller

**Reviewer 1:**
*GENERAL COMMENTS:*
*I believe the supplementary experiments that tested the pyrolyzer up to 1000 °C are of great benefit to this manuscript as it provide the necessary evidence that it is likely the responsible party for the higher recoveries. While a safe and operational 1000 °C pyrolyzer would have been ideal for all the experiments I can understand concerns if it was not entirely stable or safe for on-going operation.*

*I will now accept the manuscript for publication if the following minor corrections and updates are made:*

*SPECIFIC COMMENTS:*
*Line 29: Please write the actual value (127%) and not >100%*

We have added the actual value to this sentence.

*Lines 30-32: Please rephrase this sentence to:*
*"The low HgBr2 breakthrough on the downstream CEMs (<1%), suggest that the elevated recoveries are more likely related to sub-optimal pyrolyzer conditions or inefficient collection on the Tekran 2537 gold trap."*
*The low breakthrough is what suggests the pyrolyzer likely being responsible for the elevated recoveries, not the elevated recoveries suggesting poor pyrolyzer conditions.*

We have reworded this sentence using the reviewer's suggested sentence.

*Lines 38-39: The authors have now removed any and all discussion on the concerns of using the more general terminology RM, which was my concern in the original round of reviews "More*

*caution should be used in the definition of the term reactive mercury (RM)."*
*The "Often" (or perhaps sometimes) use of RM is by one or maybe two research groups. This does not qualify as often.*
*Since the authors are struggling for a descriptive terminology due to their own interchanging use of GOM and RM in their literature I suggest the following simple and clear addition:*
*"While the term RM does dilute some specific information in regard to the state of oxidized Hg in the atmosphere it does remove some of uncertainty as whether or not PBM contributes to the Hg collected by the CEMs."*

This qualifying statement has been added.

*Lines 117-118: Should be "We attempted to explain…"*

The error has been corrected.

*Lines 200-202: Please do not use "pretty" in a scientific paper. It is a colloquial and very unspecific term. Can the authors be more quantitative here? Use the data form SI Figure 4. What was the collection efficiency compared to 1000 °C? State this. This doesn't mean this value describes the exact inefficiency of the existing system (which should be mentioned), but it does a better job than "pretty".*

A more quantitative discussion of the pyrolyzer efficiency has been added.

*Lines 276-280: Linearity of the first filters remains poorly addressed. Fit both a linear and exponential curve to the data with equations and r2 values (as suggested below). The authors say in their response:*
*"There are only five data points, so we feel any assertion about this relationship being linear or exponential is not robust."*
*The authors seems to be using this as some form excuse as to why this should not be discussed, but on the contrary why did this again not sound some alarm bells and why was this set-up not re-tested? This goes specifically with the sentiments of myself and one other reviewer that experiments were rushed and less conclusive than they should have been. The authors are happy to conclude the linearity of the relationship based on limited data, but seem to not want to make conclusions if they pose a problem. This is and has been a concern of mine because if it is a exponential relationship then it would pose a problem for lower concentrations. Some form of description or discussion on this point IS necessary.*

Linear and exponential models equations and $r^2$ values have been added to SI Figure 5 for both sets of membranes, and more discussion has been added to the end of section 3.1.

*Line 369-371: This sentence should be changed to:*
*"Although further work is required to more definitively determine detection and quantification limits of the various CEM methodologies, based on the mean total Hg mass of 50 ± 20 pg observed in this study, the artifact of GEM uptake to the CEMs would be below the detection limit observed here."*
*This sentence finally provides an (albeit subtle) mention that work is still required on defining*

*detection and quantification limits of the CEM methodologies that I have suggested in all rounds of review.*

The requested revision has been made.

*Line 406: Please add the following after the sentence concluding on line 406:*
*"Evidence for this conclusion can be seen in by the XX % increase in Hg collection by the pyrolyzer at 1000 °C (see SI Figure 4) in supplementary experiments."*

The sentence has been added.

*SUPPLEMENTAL*
*In the current supplemental information that caption to SI Figure 5 has a caption. This caption should obviously be removed for publication.*
*BUT, it makes a perfect point. Add both linear and exponential curves with their appropriate equations and r2 values. This is something I have been inquiring about throughout the reviews and even co-authors have asked for its inclusion.*

The linear and exponential curve equations and $r^2$ values have been added to the Figure, and the mentioned comment on the figure caption has been removed.

[revised manuscript text omitted]